# Time-Aware Language Modeling for Historical Text Dating

**Han Ren, Hai Wang, Yajie Zhao** and **Yafeng Ren** *
Guangdong University of Foreign Studies, China
{hanren,wanghai,zhaoyajie}@gdufs.edu.cn
renyafeng@whu.edu.cn

## Abstract

Automatic text dating(ATD) is a challenging task since explicit temporal mentions usually do not appear in texts. Existing state-of-the-art approaches learn word representations via language models, whereas most of them ignore diachronic change of words, which may affect the efforts of text modeling. Meanwhile, few of them consider text modeling for long diachronic documents. In this paper, we present a time-aware language model named TALM, to learn temporal word representations by transferring language models of general domains to those of time-specific ones. We also build a hierarchical modeling approach to represent diachronic documents by encoding them with temporal word representations. Experiments on a Chinese diachronic corpus show that our model effectively captures implicit temporal information of words, and outperforms state-of-the-art approaches in historical text dating as well. Our code is available at: https://github.com/coderlihong/text-dating.

## 1 Introduction

The temporal dimension texts is critical to many natural language processing(NLP) tasks, such as information retrieval (Kanhabua and Nørvåg, 2016), question answering(Shang et al., 2022; Stricker, 2021), text summarization(Cao and Wang, 2022; Martschat and Markert, 2018), event detection(Sprugnoli and Tonelli, 2019), and sentiment analysis(Ren et al., 2016). Timestamps of documents provide essential clues for understanding and reasoning in time-sensitive tasks, whereas they are not always available(Chambers, 2012). One way to solve this problem is to predict when a document was written according to its content, which is also known as automatic text dating(Dalli, 2006).

Text dating has been widely introduced in computational sociology and digital humanities studies. One typical application of it is to date his-

---

*Corresponding author.

torical documents for the construction of digital libraries(Baledent et al., 2020). Such task is also called historical text dating(Boldsen and Wahlberg, 2021), diachronic text evaluation(Popescu and Strapparava, 2015), or period classification(Tian and Kübler, 2021). Compared to other dating tasks, historical text dating is more challenging as explicit temporal mentions(e.g., time expressions) that help to determine the written date of a document usually do not appear in it(Toner and Han, 2019). To solve it, current research on historical text dating focuses on document modeling, trying to find the relationship between time and linguistic features(Boldsen and Wahlberg, 2021).

There are two main issues to historical text modeling. One is to learn word representations by diachronic documents. Current research on word representation either learn static word embedding throughout the corpus(Liebeskind and Liebeskind, 2020; Yu and Huangfu, 2019), or learn dynamic word representations using pre-trained models(Tian and Kübler, 2021). However, neither of them takes into account the relation between time and word meaning. For example, *broadcast* usually refers to sowing seeds before the 20th century; after that, it means transmitting by radios or TVs in most cases. Studies on language evolution help to find the relationship between the same words in different time periods, since they often discuss them by mapping them into a same semantic space(Ferri et al., 2018). However, how to apply such methods into document modeling for historical text dating is still unexplored.

Another is document modeling for historical texts. Initial work on neural network-based document modeling employ convolutional neural networks(CNN) or recurrent neural networks(RNN) (Liebeskind and Liebeskind, 2020; Yu and Huangfu, 2019), while recent research turns to pre-trained models like BERT(Tian and Kübler, 2021) or RoBERTa(Li et al., 2022). However, these

studies always treat time as a prediction target, but not a variable in modeling, which does not help to capture the temporal characteristics of diachronic documents. In fact, evidences from the research on language modeling show that time-based models help to capture semantic change of language, and consequently improve the performance of time-sensitive tasks(Agarwal and Nenkova, 2022; Rosin et al., 2022; Röttger and Pierrehumbert, 2021). To this end, some studies attempt to incorporate the time factor into language modeling, e.g., learning explicit temporal expressions via language models(Dhingra et al., 2022; Rosin et al., 2022). However, as mentioned above, these methods are not suitable for historical text dating due to the lacking of explicit time expressions in historical texts.

In this paper, we present a time-aware language model named TALM, trying to introduce the time factor into the modeling procedure for historical text dating. Inspired by the work in language evolution, we propose to learn word representations over documents of different time periods separately. In this way, each word has time-specific variants to the vocabulary. It is important because a document should be represented by word embeddings that are temporally consistent with it. We also apply an alignment approach to map all the time-specific variants of a word into the same semantic space in order to make them comparable. In particular, we propose temporal adaptation, a representation learning approach to learn word representations having consistent time periods with documents they belong to. This approach attempts to learn temporal word representations based on the knowledge of two aspects: time-specific variants of words and their contexts, which depict the temporal attribute of words from multiple perspectives. We also build a hierarchical model for long document modeling, where the temporal adaptation is applied for word representation. We validate our model on a large-scale Chinese diachronic corpus and an English diachronic corpus. Experimental results show that our model effectively captures implicit temporal information of words, and outperforms state-of-the-art approaches in historical text dating as well. Our contributions can be summarized as follows:

- We propose a temporal adaptation approach that enables the word representation models to capture both temporal and contextualized information by learning the distributed representations of diachronic documents;

- We propose a time-aware language model for historical texts, which uses the temporal adaptation approach to obtain time-specific variants of words that are temporally consistent with the documents they belong to, thereby improving the ability to model diachronic documents;

- We report the superior performances of our model compared to the state-of-the-art models in the historical text dating task, showing the effectiveness of our model in capturing implicit temporal information of words.

## 2 Related Work

Automatic text dating follows the research roadmap from traditional machine learning to deep learning technologies, like many other NLP tasks. Early studies employ features by manual work to recognize temporal expressions within documents(Dalli, 2006; Kanhabua and Nørvåg, 2016; Niculae et al., 2014), which suffer from the problem of generalization and coverage rate. Traditional machine learning methods focus on statistical features and learning models, such as Naïve Bayes(Boldsen and Wahlberg, 2021), SVM(Garcia-Fernandez et al., 2011) and Random Forests(Ciobanu et al., 2013). Recent studies turn to deep learning methods, and the experiments show their superior performances compared to traditional machine learning ones(Kulkarni et al., 2018; Liebeskind and Liebeskind, 2020; Yu and Huangfu, 2019; Ren et al., 2022). Pre-trained models are also leveraged to represent texts for the dating task, such as Sentence-BERT(Massidda, 2020; Tian and Kübler, 2021) and RoBERTa(Li et al., 2022). Pre-trained models show state-of-the-art performances on the text dating task; however, few of them consider the time attribute of words.

Language evolution studies explore the issue by modeling words from different time periods. Such work can be categorized into three classes: 1)learning word embeddings for each time period separately, then mapping them into the same space via alignment methods(Alvarez-Melis and Jaakkola, 2018; Hamilton et al., 2016); 2) learning word embeddings for a time period first, then using them as the initial values to train word embedding for other time periods(Di Carlo et al., 2019); 3) learning unified word embeddings by introducing temporal variables into the learning procedure(Tang, 2018; Yao et al., 2018). However, static word embed-

ding has the drawback of dealing with polysemous words (Asudani et al., 2023), which may not be suitable for temporal word representation learning. In addition, there are still few studies on the language evolution modeling for historical text dating.

On the other hand, research on the temporal pretrained models shows that encoding the temporal information in language modeling is beneficial in both upstream and downstream tasks(Agarwal and Nenkova, 2022; Röttger and Pierrehumbert, 2021). Efforts have been made to build models that can learn temporal information, such as incorporating timestamps of texts into the input of the model (Pokrywka and Graliński, 2022; Pramanick et al., 2021), prefixing the input with a temporal representation(Dhingra et al., 2022; Chang et al., 2021), or learning the masked temporal information(Rosin et al., 2022; Su et al., 2022). However, there is still insufficient discussion of document modeling without explicit temporal mentions.

## 3   The Approach

In this section, we explain the mechanism of the time-aware language modeling approach based on temporal adaptation, and how it is applied to the task of historical text dating. Figure 1 shows the overall architecture of the proposed model, which consists of three main modules: word representation learning and alignment, temporal adaptation, and diachronic document modeling.

### 3.1   Word Representation Learning and Alignment

Word representation alignment is a pipeline, where word representations are learned firstly in an unsupervised manner, and then are aligned in the same semantic space. Note that each model for learning word representations is trained respectively on the documents of every time period. In our method we use BERT(Devlin et al., 2018) as the language model to learn. Let $D^t$ be the document collection of the $t$th time-period. Let a sentence $X^t = (x_1^t, \cdots, x_n^t)$, where $X^t \in D^t$, $n$ is its length, and $x_i^t$ is the $i$th token in $X^t$. Let $E^t = (e_1^t, \cdots, e_n^t)$, where $e_i^t$ is the distributed representation of $x_i^t$. We train the model with the masked language modeling task on each period separately, and extract the embedding layers as word representations:

$$E^t = \mathrm{BERT}(X^t) \qquad (1)$$

We follow the idea of Hamilton et al. (2016), mapping the word representations of different time periods into the same semantic space by orthogonal Procrustes algorithm, which aims to make them comparable. Let $W^t \in \mathbb{R}^{d \times |V|}$ be the word vector matrix at time period $t$. We use the singular value decomposition to solve for matrix $Q$, so that the cosine similarity between the word vector matrices from different time periods remains unchanged while minimizing the distance in the vector space. The objective for alignment is shown below:

$$R^t = \arg\min \left\| W^T Q - W^{t+1} \right\|_F \qquad (2)$$

where $R^t \in \mathbb{R}^{d \times d}$.

### 3.2   Temporal Adaptation

The meaning of words always changes over time. Hence it is necessary to consider the model's ability to adapt different meanings of words over time. This problem is similar to domain adaptation(Ramponi and Plank, 2020), where the model should have the ability of transferring from the source semantic space domain to the target one. In the historical text dating task, the time period of the document to be predicted is not known in advance, hence the model needs to determine the word representations of a time period that are adaptive to the document. This is called temporal adaptation in our model. This subsection shows the temporal adaptation approach. The middle part in Figure 1 represents the temporal adaptation module proposed in this paper. The input of this module are documents, which consist of segmented blocks. We encode each word with three embeddings: token embedding, position embedding, and block embedding. The block embedding of a word is utilized to indicate the sequential information of the sentence it positioned. The word embedding is defined as follows:

$$E_{input} = E_{token} + E_{position} + E_{block} \qquad (3)$$

The main part of the temporal adaptation module is a Transformer Encoder. $h_i$ represents the hidden space vector for each token obtained from the Transformer Encoder. One of the purposes of this module is to adapt $h_i$ to $\overline{e_i}^t$, which can make the hidden representations fit temporal domain.

On the other hand, contexts also indicate temporal information, which means modeling contextual information helps to determine the time period a text may belongs to. To this end, we design a

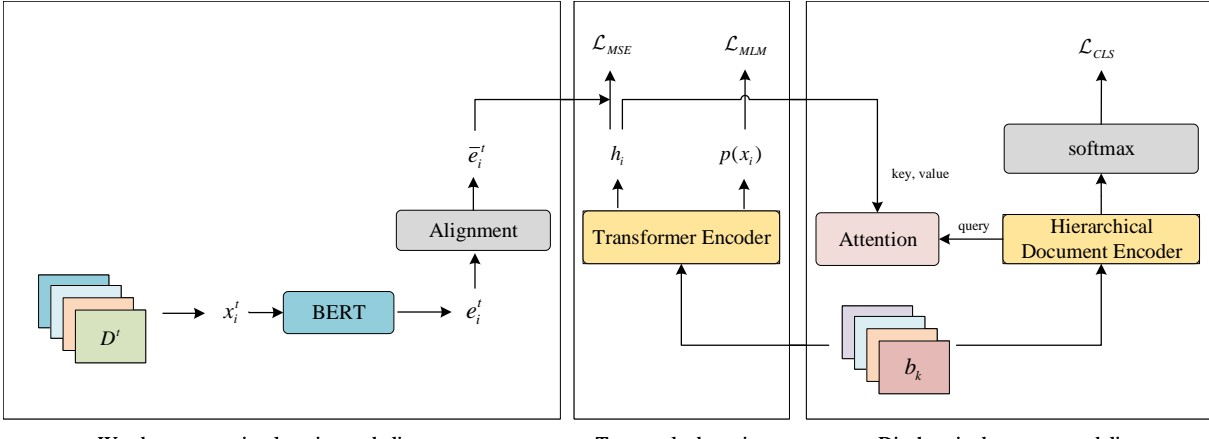

Figure 1: The architecture of our model. The left part is the word representation and alignment module, which is to learn and align the temporal word representation in each time period; the middle part is the temporal adaptation module, which aims to learn the time-specific variants of words and their contextual information; the right part is the hierarchical modal for diachronic documents, which incorporate the results of temporal adaptation into language modeling.

masked language modeling task to learn the contextual information of the input texts. Different from other temporal masked language models, we do not consider the idea of masking temporal information as they did not obviously appear in the text. Alternately, we use a fundamental method by masking a portion of words randomly, based on the assumption that the contexts of word also indicate the temporal information since they are in the same time period. Hence we follow the masked language modeling task in BERT training process.

### 3.3 Diachronic document modeling

The aim of diachronic document modeling is to obtain a document-level representation for long historical texts. The novelty of this approach lies in the combination of the hierarchical document modeling and temporal adaptation. By doing so, temporal features are incorporated into each layer of the transformer encoder, allowing the hierarchical model to learn implicit temporal information according to the knowledge gained from temporal adaptation. Experiments show that this method improves the performance of the text dating task.

Inspired by (Grail et al., 2021), we propose a hierarchical model for long diachronic document modeling. The hierarchical structure in this paper consists of a word transformer and a block transformer. The word transformer is responsible for learning local token representations in each block, while the block transformer learns the contextual information of blocks. Since transformer encoder

on the block encoding layer lacks the sequential information of the blocks, we employ three types of embedding to indicate the word order information and block order information, as mentioned in equation 3. After block transformer encoding, the model obtains the global block-aware representations, and put it to the next layer for further learning.

In order to make use of the temporal representations of words in diachronic document modeling, we explore the bridging approaches within the Transformer Encoder layer to incorporate the temporal feature representation during the hierarchical modeling process. Specifically, three bridging approaches are built:

- **Feature Aggregation** This approach directly adds the input at each layer with the temporal representations of words.

- **Dot Attention** The document input representation is used as the query, while the time feature representations of the words are employed as the key and value. The dot product attention mechanism is applied to each input sample to integrate the temporal information.

- **Additive Attention** Similar to dot attention, an additional attention parameter matrix is introduced in this method. The document input representation serves as the query, while the time feature representations of the words are utilized as the key and value. Using the additive attention mechanism, the temporal feature

information is weighted and merged into the original input at each step.

After N-layer encoding, we utilize an attention mechanism to determine the importance of different document blocks. Attention scores are calculated to represent the importance of each document block, and these scores are utilized to weigh the feature information of each block. Finally, we obtain the representation of the document, which is then evaluated for historical text dating.

### 3.4 Training objective

The overall training objective is a combination of the losses from two modules: temporal adaptation and diachronic document modeling.

For temporal adaptation module, we have two training objectives in learning process. One of the learning objective is that transforming hidden representation learned by Transformer Encoder into target temporal domain, which means minimize the distance between $h_i$ and $\bar{e}_i^t$. Hence we adopt mean squared error as our loss function, which is defined below:

$$\mathcal{L}_{MSE} = \frac{1}{N} \sum_{i=1}^{N} MSE(\bar{e}_i^t, h_i) \qquad (4)$$

where $N$ is the number of tokens. By doing so, the model can map the word representations of the input text to the semantic space domain of the corresponding time period, making the model to be represented more appropriate.

Another training objective of temporal adaptation module is masked language modeling(MLM), which aims to maximize the predict probability of masked words. Let $X_\Pi = \{\pi_1, \pi_2, \ldots, \pi_K\}$, which denote the indexes of the masked tokens in the sentence $X$, where $K$ is the number of masked tokens. Denote that $X_\Pi$ is the collection of masked tokens in sentence X, and $X_{-\Pi}$ is the collection of unmasked tokens. The learning objective of MLM is defined as:

$$\mathcal{L}_{\text{MLM}} = \frac{1}{K} \sum_{k=1}^{K} \log p(x_{\pi_k}|X_{-\Pi}; \theta). \qquad (5)$$

For diachronic document modeling module, we use the cross-entropy loss as the training objective for historical text dating task. In this loss function, $y_{i,c} \in \{0, 1\}$, representing whether the true class of document $i$ belonging to class $c$, and $C$ is the number of classes. The cross-entropy computes as:

$$\mathcal{L}_{CLS} = \frac{1}{N} \sum_{i=1}^{N} \sum_{c=1}^{C} y_{i,c} \log p_{i,c} \qquad (6)$$

Finally, We aggregate three losses to form the mixture training objectives of our model. The overall training objective is formulated as the sum of the above three losses:

$$\mathcal{L} = \mathcal{L}_{CLS} + \mathcal{L}_{MSE} + \mathcal{L}_{MLM} \qquad (7)$$

where $\mathcal{L}_{CLS}$ represents the loss function of text dating,both $\mathcal{L}_{MSE}$ and $\mathcal{L}_{MLM}$ are the loss function of temporal adaptation.

## 4 Experiments

This section shows the experimental results as well as a discussion of the proposed model. Specifically, we give the details about the experimental settings first, then we introduce the datasets and the preprocessing methods for them. In subsection 4.3, we show the results of the comparative experiments, the ablation tests and the parameter evaluations. Finally, a discussion of our model is given.

### 4.1 Dataset

Our experiments are conducted on two datasets. One is a Chinese dataset including documents of Chinese official dynastic histories(a.k.a. Twenty-Four Histories) with the time spanning from 2500 B.C. to 1600 A.D., and the other is an English dataset named Royal Society Corpus(Kermes et al., 2016), with the time spanning from 1660 to 1880. Such kind of datasets have a long time span, so that they are suitable for our model to explore the performance of temporal modeling. Considering that some historical records in the Twenty-Four Histories Corpus were not written in its time periods but compiled and summarized by the later writers, we assign a timestamp to each document of the corpus based on the time it is finished. We divided the corpus into smaller blocks for the convenience of the analysis. Specifically, we extract every thirty sentences from the original corpus as one block, with each sentence containing a maximum of thirty Chinese characters or English words. These samples are labeled with their corresponding time periods and randomly split into training, validation, and testing sets in an 8:1:1 ratio, respectively. Statistical results of the dataset is shown in Table 1.

| Dataset | Year of publication | Blocks | Tokens | Sentences |
|---|---|---|---|---|
| Twenty-Four Histories Corpus | Western Han | 921 | 557204 | 27630 |
| | Eastern Han | 1135 | 432991 | 34050 |
| | Western Jin | 975 | 502003 | 29250 |
| | Southern Song | 1151 | 560857 | 34530 |
| | Southern Liang | 1871 | 919178 | 56130 |
| | Northern Qi | 1826 | 869476 | 54780 |
| | Tang | 8098 | 3978260 | 242940 |
| | Later Jin | 3104 | 1566592 | 93120 |
| | Song | 3847 | 1888350 | 115410 |
| | Yuan | 7839 | 3763214 | 235180 |
| | Ming | 2141 | 1033055 | 64230 |
| | Qing | 5242 | 2439673 | 157260 |
| Royal Society Corpus | 1660-1680 | 2495 | 741897 | 74850 |
| | 1680-1700 | 2643 | 755797 | 79290 |
| | 1700-1720 | 3048 | 732179 | 91440 |
| | 1720-1740 | 2586 | 676415 | 775801 |
| | 1740-1760 | 4524 | 1223036 | 135720 |
| | 1760-1780 | 4846 | 1458099 | 145380 |
| | 1780-1800 | 4128 | 1444720 | 123840 |
| | 1800-1820 | 5590 | 1684756 | 167700 |
| | 1820-1840 | 6769 | 2111766 | 203070 |
| | 1840-1860 | 10032 | 3174480 | 300960 |
| | 1860-1880 | 9061 | 2870448 | 271830 |

Table 1: Statistical results of the dataset

## 4.2 Experiment Result

**Baselines** We conduct experiments on two Histories dataset using our proposed method. All the parameters are randomly initialized and tuned during the training process. We compare our method with the state-of-the-art models.

- **HAN**(Yang et al., 2016): Hierarchical Attention Network combines word attention and sentence attention mechanisms to obtain document-level representation.

- **DPCNN**(Johnson and Zhang, 2017): Deep pyramid convolutional network is a word-level deep convolutional neural network. By using region embedding and residual connection, it can promote network capturing the long-distance dependence between words.

- **BERT**(Devlin et al., 2018): The pre-trained BERT-base model and we finetuned it with the training data for classification.

- **Longformer**(Beltagy et al., 2020): Longformer utilizes a self-attention model by using global and sliding window information to model long sequences.

- **Hierarchical BERT**(Khandve et al., 2022): Hierarchical BERT uses BERT as the hierarchical text encoder, and leverage LSTM to encode the sentence feature to obtain the final representation of the document.

- **SentenceBert** (Tian and Kübler, 2021): SentenceBERT uses Siamese and triple network architectures to generate sentence representa-

tions.

- **RoBERTa**(Li et al., 2022): RoBERTa is an improved version of BERT, which uses a larger training corpus and a dynamic mask method to obtain a better representation of the text.

**Parameter Setting** During the training stage, the input contains 30 sentences, each of which contains 30 words. The dimension of the word vector is 768, and both the temporal adaptation and the hierarchical model has 6 layers, and the dropout rate is 0.1. The batch size of the training is 8 and the learning rate is 1e-5. The optimizer we use is AdamW. All the systems are conducted over 5-fold cross validation.

**Performance Comparison** We compare our proposed model TALM with other baseline models, and the performance results of each model are shown in Table 2. We use macro precision, macro recall and macro F1 as our metrics. The best-performing values are in bold format.

As shown in Table 2, on the Twenty-Four Histories Corpus, our proposed model achieves the best performance in the text dating task with an F1 score of 84.99%, outperforming other baseline models. We can see that, the methods based on the structure of pre-trained language models outperform the traditional neural network-based approaches. Longformer achieves a best F1 score of 81.6% in the baseline models, which is specifically used to handle the problem of long text input. However, it is challenging to incorporate temporal information into the Longformer model, as common attention mechanisms have high computation complexity for long document inputs. Hence, in this study, we adopt a hierarchical document modeling structure, which reduces the computational complexity for long input sequence. Among the baseline models, both HAN and Hierarchical BERT utilize hierarchical document structures and achieve competitive performance. Our proposed model leverage temporal adaptation and bridging structures in learning process, outperforms these hierarchical models by 6.9% and 3.39% in F1 score respectively, demonstrating the effectiveness of incorporating time information in text dating tasks.

In the methods used for dating, SBERT and RoBERTa achieve similar model performance, while LSTM performs a relatively low performance, due to its limited modeling capability. However, these baseline models primarily focus on the

language features of the text itself and do not incorporate the temporal information associated with the text. In contrast, our proposed model incorporates the corresponding temporal information of the input through different mechanisms. The additive attention achieves the best performance, outperforming SBERT and RoBERTa by 4.28% and 4.24% in F1 score, respectively, demonstrating the advantage of TALM in learning temporal information.

On the Royal Society Corpus, our model achieved the best performance among all the models, similar to that on the Chinese corpus. Specifically, our model with additive attention gains an increasing 2.58% F-1 score compared with the best baseline SBERT. It suggests that our model helps to learn temporal information and incorporate this information effectively into text dating tasks in both Chinese and English texts.

**Comparison Of Bridging Methods** We use different bridging approaches to make the model having the ability of time-awareness. Here we will discuss the impact of these approaches to model performances. As shown in Table 2, each of these approaches leads to an improving performance, showing the effectiveness of integrating temporal information through bridging mechanisms. Overall, the additive attention approach achieves the best results. This attention mechanism introduces additional attention matrix parameters, which control the amount of temporal word representation information selected during training. As a result, the model can learn word vector representations corresponding to specific time periods more effectively.

**Flexible Evaluation Criteria** This study focuses on the historical text dating task. However, in the historical corpora, word meaning changes in a gradual way, and the semantic similarity between texts from adjacent periods may be higher than those that have longer time span. Inspired by (Li et al., 2022), we define a more flexible evaluation metric called $Acc@K$. This metric treats the prediction result adjacent to the correct time period as correct one, which is a more flexible metric. $N_{acc}@K$ represents the number of the correct cases, where those having a period class distance of $\pm \lfloor \frac{K}{2} \rfloor$ are also taken into account. $N_{all}$ represents the total number of samples in the dataset. The specific mathematical formula is defined as follows:

$$Acc@K = \frac{N_{acc}@K}{N_{all}} \qquad (8)$$

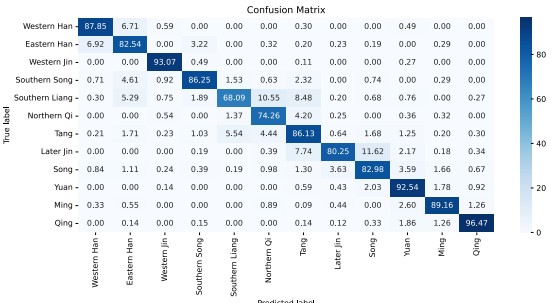

Figure 2: The confusion matrix of our model on the Twenty-Four Histories Corpus.

We use the metric of $Acc$, $Acc@3$ and $Acc@5$ to evaluate our model, As can be seen from Table 2 that: 1) our model outperforms the baselines in most cases, showing its robustness in both rigid and relaxed evaluation criteria; 2) our model can better distinguish cases of adjacent time periods, compared with the other baselines.

**Ablation Study** In the ablation experiments, we analyze the contributions of these components by evaluating TALM without time-specific variants of words, TALM without context learning, and TALM without temporal attention.

When the temporal attention module is removed, the performance of the bridging module is also eliminated. Therefore, temporal adaptation and time-aware language modeling cannot be integrated with the dating task. This model performs the worst with an F1 score of 76.69%. When the time-specific variants of words are removed, the model does not work well to learn temporal word representations, leading to a performance decrease of 3.42% in F1 score. Similarly, when the context learning module is removed, the model does not work well to learn contextual information for temporal word representations, resulting in a performance decrease of 6.31% in F1 score. These results indicate that all the module play crucial roles in the text dating task.

### 4.3 Analysis

**Case Study** In this section, we discuss the performances of our model in different time periods, analyze cases that our model fail to recognize, and investigate the impact of the temporal adaptation module in a visualization way. Figure 2 shows the confusion matrix of our model on the test set. We can see that it is much difficult to predict the time period of the Southern Liang. Actually, most samples are falsely predicted to the Dynasty Tang,

| Method | Twenty-Four Histories Corpus | | | | | | Royal Society Corpus | | | | | |
|---|---|---|---|---|---|---|---|---|---|---|---|---|
| | P | R | F1 | Acc | Acc@3 | Acc@5 | P | R | F1 | Acc | Acc@3 | Acc@5 |
| Han(Yang et al., 2016) | 77.09 | 79.12 | 78.09 | 81.32 | 92.23 | 96.88 | 56.84 | 56.78 | 56.45 | 59.03 | 88.17 | 94.65 |
| DPCNN(Johnson and Zhang, 2017) | 75.57 | 76.94 | 76.25 | 80.60 | 90.63 | 96.38 | 55.36 | 53.64 | 53.32 | 56.02 | 86.13 | 94.15 |
| BERT(Devlin et al., 2018) | 81.11 | 80.07 | 80.59 | 83.72 | 92.29 | **97.36** | 57.63 | 57.27 | 57.27 | 59.27 | 87.71 | 95.01 |
| SSGC(Zhu and Koniusz, 2020) | 80.76 | 81.44 | 81.10 | 82.20 | 90.05 | 97.15 | - | - | - | - | - | - |
| Longformer(Beltagy et al., 2020) | 81.83 | 81.37 | 81.60 | 82.92 | 90.34 | 95.84 | 57.93 | 58.21 | 57.41 | 59.83 | 89.00 | 95.24 |
| Hierarchical BERT(Khandve et al., 2022) | 81.59 | 79.47 | 80.16 | 82.34 | 91.12 | 95.73 | 45.52 | 45.66 | 45.41 | 47.45 | 75.90 | 86.01 |
| LSTM(Yu and Huangfu, 2019) | 68.62 | 69.36 | 68.99 | 72.89 | 89.25 | 95.80 | 54.50 | 53.99 | 53.61 | 55.62 | 88.81 | **95.32** |
| SBERT(Tian and Kübler, 2021) | 80.50 | 80.92 | 80.71 | 81.80 | 91.20 | 96.72 | 57.92 | 59.17 | 58.19 | 60.09 | 88.00 | 94.63 |
| RoBERTa(Li et al., 2022) | 80.42 | 81.08 | 80.75 | 82.23 | 91.14 | 97.31 | 57.92 | 58.05 | 57.52 | 59.71 | 88.87 | 94.94 |
| feature aggregation | 80.63 | 81.68 | 81.15 | 83.64 | 91.33 | 96.50 | 58.11 | 58.04 | 57.86 | 59.64 | 87.57 | 94.16 |
| dot attention | 82.19 | 81.87 | 82.03 | 83.96 | 92.02 | 96.82 | 56.72 | 56.74 | 56.56 | 58.86 | 87.01 | 94.25 |
| additive attention | **84.96** | **85.02** | **84.99** | **86.74** | **93.50** | 97.32 | **61.11** | **61.38** | **60.77** | **62.81** | **88.33** | 94.81 |

Table 2: Performances of our model and baselines on the Twenty-Four Histories corpus and Royal Society Corpus. The evaluation metrics include P, R, F1, Acc, Acc@3 and Acc@5.

| Model | Twenty-Four History Corpus | | | Royal Society Corpus | | |
|---|---|---|---|---|---|---|
| | P | R | F1 | P | R | F1 |
| TALM(Our method) | **84.96** | **85.02** | **84.99** | **61.11** | **61.38** | **60.77** |
| w/o time-specific variants of words | 81.60 | 81.54 | 81.57 | 58.44 | 58.39 | 58.11 |
| w/o context learning | 78.73 | 78.63 | 78.68 | 53.74 | 53.66 | 53.49 |
| w/o temporal attention | 79.62 | 73.97 | 76.69 | 51.32 | 51.28 | 50.82 |

Table 3: Results of ablation study on the Twenty-Four Histories corpus.

| Period | TALM | | | RoBERTa(Li et al., 2022) | | |
|---|---|---|---|---|---|---|
| | P | R | F1 | P | R | F1 |
| Western Han | 87.66 | 88.03 | 87.85 | 93.80 | 90.60 | 92.17 |
| Eastern Han | 77.07 | 88.85 | 82.54 | 80.34 | 91.08 | 95.37 |
| Western Jin | 90.38 | 95.92 | 93.07 | 85.45 | 95.92 | 90.38 |
| Southern Song | 89.61 | 83.13 | 86.25 | 89.86 | 74.70 | 81.58 |
| Southern Liang | 80.00 | 59.26 | 68.09 | 68.68 | 57.87 | 62.81 |
| Northern Qi | 69.06 | 80.30 | 74.26 | 54.27 | 81.81 | 65.26 |
| Tang | 83.48 | 88.96 | 86.13 | 83.70 | 83.19 | 83.45 |
| Later Jin | 92.70 | 70.75 | 80.25 | 87.44 | 54.32 | 67.01 |
| Song | 81.48 | 84.54 | 82.98 | 74.23 | 80.22 | 77.11 |
| Yuan | 89.86 | 95.38 | 92.54 | 85.63 | 92.12 | 88.76 |
| Ming | 89.40 | 88.92 | 89.16 | 68.51 | 87.03 | 76.67 |
| Qing | 96.70 | 96.25 | 96.47 | 96.52 | 93.62 | 95.05 |

Table 4: the result of macro P, R, and F1-score value on each period.

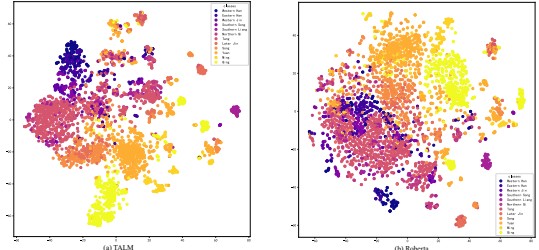

Figure 3: The document distribution representation visualization of the two models.

with a few samples predicted as the Northern Qi. This could be attributed to the relatively short time duration of each period in the Southern and Northern Dynasties. In other words, the semantic gap between these three dynasty is not much obvious. Furthermore, the Later Jin period lasted for only about ten years in the ancient China history, and this may lead to low performance in identifying if a document belongs to this time period. This indicates that the Later Jin period serves as a transitional time period with semantic and textual writing styles similar to the Tang and Song Dynasties.

Table 4 shows that our model outperforms the

state-of-the-art model RoBERTa in those cases who have a short historical time duration. Specifically, our model gains an increasing F1 score of 5.28%, 9.00% and 13.40% in the time period of Southern Liang, Northern Dynasty of Qi, Later Jin.

It also can be seen from Table 2 that, the score of the metric $Acc@5$ of our model significantly higher than that of $Acc$. It suggests that texts of adjacent periods have a great impact on the performance of the model, leading to a higher probability of the error-predicted period results close to the gold labels than to other periods. To make an in-depth analysis of such errors, we sample the false-predicted cases of Southern Liang, which got the lowest recall score on the test set. As shown in Table 5, the sentence 1-3 belong to the South-

| | Example | Interpretation | Model Result | Gold Label | Period Interval |
|---|---|---|---|---|---|
| 1 | 故我朝庭解甲，息心东南之略，是为不欲违先故之大信也。 | 所以我们朝廷停止进攻，不打算占领东南，这是不想违背先人的诺言。
Therefore, our military's decision to cease the offensive and refrain from occupying the southeastern region does not contradict the promises of our predecessors. | Northern Qi | Southern Liang | +1 |
| 2 | 驱信道之成终，表昧世之亏始。 | 奔跑在诚信道路上才快速到达了成功的终点，表现出胡里胡涂不明世事，这将是吃亏的开始。
Success can only be achieved through honesty, while being confused and ignorant of worldly matters will be the beginning of a disadvantage. | Northern Qi | Southern Liang | +1 |
| 3 | 信宿之间，宣阳底定。 | 连续两夜之间，宣阳得以平定。
After two consecutive nights, Xuanyang was pacified. | Tang | Southern Liang | +2 |
| 4 | 国子、太学、四门学生参见则服之。 | 国子、太学、四门学生拜访就需要穿戴这些服饰。
Students from Guozi, Taixue, and Four Schools need to wear these garments when visiting. | Song | Later Jin | +1 |
| 5 | 时李勣已率兵攻辽东城。 | 此时李勣已经带领军队攻辽东城。
At this time, Li Ji had already led the army to attack Liaodong City. | Song | Later Jin | +1 |
| 6 | 富贵不离其身，然后能保其社稷。 | 能够保持富有和尊贵，然后才能保住家国的安全。
To maintain wealth and nobility is to ensure the security of the homeland. | Tang | Later Jin | -1 |

Table 5: Examples of errors.

ern Liang period, whereas our model misclassifies them into the class of Northern Qi or Tang, which are adjacent periods to the gold label. Likewise, the sentence 4 and 5 are both texts from the Later Jin period, however the model misjudges them as the texts of the Song period. As a matter of fact, there is a very short time interval between the Later Jin and the Song period, nearly 13 years. The main reason of these errors may lie in the high similarity of language usage as such periods are very close, and language changes without striking difference are insufficient for more precise prediction. In other words, there is not significant change of words in these adjacent periods, making it difficult to provide discriminative features for temporal language modeling.

**Time-aware Visualization** In this section, we further investigate the impact of the temporal semantic learning module on the text dating task and compare it with the RoBERTa model. To avoid the influence of the contexts to word representations during the hierarchical document modeling process, we select the representations obtained after the first layer of additive attention in the document encoding module. After encoding with this layer of attention, the model's input incorporates the word representations with temporal information. We take the average of the word representations as the document vector and used the T-SNE method to visualize the document vectors in the test set. Similarly, we also select the word vector input layer of the RoBERTa model for comparison. Figure 3 shows the visualized results, indicating that our model has a better performance to evaluate the temporally semantic relationship among documents.

## 5 Conclusion

In this paper, we propose to incorporate the time attribute into the language modeling for histori-

cal text dating by building TALM, a time-aware language model. To address the problem of the model suffering from missing temporal expressions in historical texts, we propose a temporal adaptation model that makes the model have the ability of time-awareness. We also propose a hierarchical modeling method to incorporate the temporal adaptation and long document modeling. We conduct experiments using different attention mechanisms, showing the effectiveness of integrating temporal information on historical text dating task. Our study provides new evidences to support the viewpoint that encoding temporal information contributes to improving the performances of language modeling.

## Limitations

In this study, we make our experiments on datasets of two languages, Chinese and English. In order to further validate the effectiveness of the proposed model, datasets of more language versions are expected in order to investigate the model performance on the modeling of multiple languages. Additionally, regarding the division and definition of historical periods, this study adopted a coarse-grained labeling standard. Due to the chronological order of the corpus, coarsegrained labeling may not accurately represent the exact time of the texts. Therefore, in the future, we plan to collect or construct fine-grained textual corpora to improve the performance of temporal information learning and enhance the accuracy of the text dating task.

## Acknowledgements

This work is supported by Research Fund of National Language Commission(grant No.YB145-2), National Natural Science Foundation of China(grant No. 61977032).

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
