# OpenReview forum: "Time-Aware Language Modeling for Historical Text Dating"
_EMNLP/2023/Conference — EMNLP 2023 Findings_

### Official Review · Reviewer_8wH2 · 2023-07-26

**Typos Grammar Style And Presentation Improvements:** Line 285
**Soundness:** 2

**Excitement:**

2: Mediocre: This paper makes marginal contributions (vs non-contemporaneous work), so I would rather not see it in the conference.

**Paper Topic And Main Contributions:**

This paper studies automatic text dating (ATD) by presenting a time-aware text encoder named TALM, which is aligned and adapted to represent temporal information. The paper also proposes a diachronic document modeling framework to obtain a document-level representation for long historical texts.

The authors conduct experiments on one dataset and the proposed method demonstrates improvements over selected baselines.

**Questions For The Authors:**

Line 420-423 mentioned that "In addition, this study does not involve external language knowledge, therefore only the structural components of pre-trained language models are used without loading pre-trained model parameters". Do you mean that only the Transformer architecture is used but the parameters are randomly initialized? It seems from Figure 1 that you do need a pretrained model (BERT).

**Reasons To Accept:**

* The problem studied in the paper is interesting and not commonly seen in the NLP venues.
* The method design is reasonable.

**Reasons To Reject:**

* The novelty of the method is low. The method for aligning word representations is directly borrowed from Hamilton et al. (2016). The hierarchical document modeling module is very similar to Grail et al. (2021).
* Only one dataset is used for evaluation, which does not seem sufficient to validate the effectiveness of the method.
* (Not a major point to reject) The presentation of the paper is overall subpar. The formulas seem to be inserted as figures, and the arrangement of texts is weird here and there (e.g., the way that the numbers and names displayed in Table 2 really makes it hard to read). The authors should consider using LaTex template for formatting.

**Reproducibility:**

3: Could reproduce the results with some difficulty. The settings of parameters are underspecified or subjectively determined; the training/evaluation data are not widely available.

**Reviewer Confidence:**

3: Pretty sure, but there's a chance I missed something. Although I have a good feel for this area in general, I did not carefully check the paper's details, e.g., the math, experimental design, or novelty.

---

> ### Author Rebuttal · Authors · 2023-08-28
>
> Q1: The novelty of the method is low. The method for aligning word representations is directly borrowed from Hamilton et al. (2016). The hierarchical document modeling module is very similar to Grail et al. (2021).
>
> A1: There are two main contributions in this paper, as stated in the first section. One is that we propose an approach to learn temporally consistent word representations with those documents they belong to, where both temporal and contextualized information of words are captured to build distributed representations for diachronic document modeling. Here we employ a word representation alignment algorithm(Hamilton et al.) to map words of different time periods into the same semantic space, to reduce the learning cost. But the alignment itself is neither the main contribution nor the key point in this paper.
>
> The other is that we propose a time-aware language model for historical texts, where the model can automatically find the appropriate time-specific variant of a word that is temporally consistent with the document it belongs to. Here we build a hierarchical method to model historical documents, which is also applied in many NLP tasks to model long documents. But the hierarchical method itself is neither the main contribution nor the key point in this paper.
>
> Q2: Only one dataset is used for evaluation, which does not seem sufficient to validate the effectiveness of the method.
>
> A2: Thanks for your comments. We will validate our method on more datasets and report it in the revised manuscript.
>
> Q3: (Not a major point to reject) The presentation of the paper is overall subpar. The formulas seem to be inserted as figures, and the arrangement of texts is weird here and there (e.g., the way that the numbers and names displayed in Table 2 really makes it hard to read). The authors should consider using LaTex template for formatting.
>
> A3: Thanks for your comments. We will re-format our manuscript with the latex template.
>
> Q4: Line 420-423 mentioned that "In addition, this study does not involve external language knowledge, therefore only the structural components of pre-trained language models are used without loading pre-trained model parameters". Do you mean that only the Transformer architecture is used but the parameters are randomly initialized? It seems from Figure 1 that you do need a pretrained model (BERT).
>
> A4: "In addition, this study does not involve external language knowledge, therefore only the structural components of pre-trained language models are used without loading pre-trained model parameters". It means that the pre-trained models for performance comparison, such as BERT and Hierarchical BERT, are trained only by using the dataset built in this paper, and the aim is to explore the contribution of the temporal adaptation method in a fair environment. The BERT in Figure 1 is not a comparison system. We use it to get word representations by conducting a self-supervised learning procedure.

---

### Official Review · Reviewer_KmTN · 2023-08-05

**Soundness:** 4

**Excitement:**

4: Strong: This paper deepens the understanding of some phenomenon or lowers the barriers to an existing research direction.

**Paper Topic And Main Contributions:**

Automatic text dating is to predict when a document was written based on its content. This work address the existing works' limitation in capturing time-aware word and document representations with language modeling apporach.

To be specific, it propose to learn a time-aware and context-aware word representation and use it to hierarchically build document representation. The authors conduct experiments on Chinese historical corpus, showing a good improvement comparing with existing method in this task, showing the effectiveness of the proposed method.

**Questions For The Authors:**

I have several questions concerning both the method and experiment parts, hoping the author can clarify or make revisions in subsequent versions to enhance the clarity for future readers:

1. What is Eblock in Eq(1) and how it is represented?

2. In Figure (1), how is the learned h_i is used in document modeling? How are the q, k, v built? What is b_k?  By saying "temporal features are incorporated into each layer" in Section 3.3, do you mean h_i?

3. In ablation study, by "w/o" temporal attention, do you mean remove L_CLS in Eq(6)? If that is the case, how do you generate the prediction y with removing the classification output? If not, can you be more specific what does this w/o mean?

4. In Parameter setting, it's mentioned that training contains 30 sentences while this is not aligned with the dataset showed in Table 1. What are the training sentences here?

5. How is the weights used in 3.1 initialized?


**Reasons To Accept:**

1. The task studied in this work is interesting and is useful for researchers seeking a deeper understanding of the history of humanity.

2. It raises valuable discussion on the preservation of time-sensitive information within the framework of language modeling.

3. The proposed method can address the raised issues well, and also showing a good improvement compared to existing methods.

**Reasons To Reject:**

1. The study, along with the employed baseline comparisons, centers exclusively around Chinese corpus, thereby confining the applicability of the proposed method.

2. The absence of a comparison with Large Language Models such as the GPT models or Chinese Language Models might be a concern.

**Reproducibility:**

3: Could reproduce the results with some difficulty. The settings of parameters are underspecified or subjectively determined; the training/evaluation data are not widely available.

**Reviewer Confidence:**

4: Quite sure. I tried to check the important points carefully. It's unlikely, though conceivable, that I missed something that should affect my ratings.

**Typos Grammar Style And Presentation Improvements:**


I also have several revise suggestions for the author to consider:

1. I understand the limitation of this work is focusing on Chinese historical text dating. It would be beneficial to expand the conversation to encompass potential applications in other languages and provide insights based on the distinct characteristics of each language.

2. Readers will be interested in the public LLM-based performance which may also be considered as baselines.

3. For the representation learned in Section 3.1, by splitting the data into different collections, it's making the data more sparse.  I wonder if it is a better way to pretrain the model on the entire dataset, followed by a subsequent training within each collection.

4. Still for section 3.1, my understanding of the motivation of Eq(2) is to model language evolution. It will be good if you can add some explanation of the motivation behind this.

5.  You can try refine table 4 to save some space.

6. Again, to mitigate potential ambiguities upon reading this submission, I suggest to refine the content or add specific details based on the details provided in my earlier question section, especially in cases where clarity indeed is lacking.

---

> ### Author Rebuttal · Authors · 2023-08-28
>
> Q1: The study, along with the employed baseline comparisons, centers exclusively around Chinese corpus, thereby confining the applicability of the proposed method.
>
> A1: Thanks for your comment. We will validate the applicability of the proposed method on more historical text datasets and report it in the revised manuscript.
>
> Q2: The absence of a comparison with Large Language Models such as the GPT models or Chinese Language Models might be a concern.
>
> A2: Thanks for your comment. We will report supplement experiments of large language models in our revised manuscript.
>
> Q3: What is Eblock in Eq(1) and how it is represented?
>
> A3: Eblock denotes the position of the sentence in a block. It indicates the sentence order.
>
> Q4: In Figure (1), how is the learned h_i is used in document modeling? How are the q, k, v built? What is b_k? By saying "temporal features are incorporated into each layer" in Section 3.3, do you mean h_i?
>
> A4:
> 1) "how is the learned h_i used in document modeling?"
> Attention models are utilized to incorporate temporal features(h_i) into word representations, and the aim is to enable the time-aware ability of document modeling by using word representations with temporal information.
> 2) "How are the q, k, v built?"
> q denotes the word representation by hierarchical document encoding; k, v denotes h_i. The attention model is to incorporate temporal information into hierarchical document representation.
> 3) "What is b_k?"
> b_k denotes the kth block of the dataset. In hierarchical modeling, long documents are usually partitioned into multiple blocks. In this paper, each block includes 30 sentences.
> 4) "By saying "temporal features are incorporated into each layer" in Section 3.3, do you mean h_i?"
> Yes. We incorporate the weighted h_i from the attention model into each layer of the document encoder.
>
> Q5: In ablation study, by "w/o" temporal attention, do you mean remove L_CLS in Eq(6)? If that is the case, how do you generate the prediction y with removing the classification output? If not, can you be more specific what does this w/o mean?
>
> A5: w/o temporal attention means that the attention module in the diachronic document modeling stage in Figure 1 is removed. In other words, it only remains a hierarchical document encoder in the stage. The aim of this ablation test is to explore the effectiveness of the temporal adaptation model.
>
> Q6: In Parameter setting, it's mentioned that training contains 30 sentences while this is not aligned with the dataset showed in Table 1. What are the training sentences here?
>
> A6: Sentences in Table 1 refer to the total amount of sentences in the document of the same time period. While in the training stage, each sample(block) contains 30 sentences.
>
> Q7: How is the weights used in 3.1 initialized?
>
> A7: The weights used in section 3.1 are initialized randomly.
>
> Q8: I understand the limitation of this work is focusing on Chinese historical text dating. It would be beneficial to expand the conversation to encompass potential applications in other languages and provide insights based on the distinct characteristics of each language.
>
> A8: Thanks for your comment. We will continuously consider more application scenarios of our model. We believe that it helps to model more time-sensitive tasks, but not only in the historical text dating task.
>
> Q9: Readers will be interested in the public LLM-based performance which may also be considered as baselines.
>
> A9: Thanks for your comment. We will report supplement experiments of large language models in our revised manuscript.
>
> Q10: For the representation learned in Section 3.1, by splitting the data into different collections, it's making the data more sparse. I wonder if it is a better way to pretrain the model on the entire dataset, followed by a subsequent training within each collection.
>
> A10: Thanks for your comment. The idea of splitting the data into different periods is inspired by the research on lexical evolution, such as (Hamilton et al., 2016; Chen et al., 2022), and the aim of it is to model more precise temporal information of words. But we are also interested in your suggestion. In our future work, we will explore the method of training our model over the entire dataset.
>
> Q11: Still for section 3.1, my understanding of the motivation of Eq(2) is to model language evolution. It will be good if you can add some explanation of the motivation behind this.
>
> A11: Eq(2) denotes a learning objective to minimize the distance between two vectors by solving the matrix Q via SVD.
>
> Q12: You can try refine table 4 to save some space.
>
> A12: Thanks for your comment. We will re-forrmat the manuscript.
>
> Q13: Again, to mitigate potential ambiguities upon reading this submission, I suggest to refine the content or add specific details based on the details provided in my earlier question section, especially in cases where clarity indeed is lacking.
>
> A13: Thanks for your comment. We will follow your suggestions to revise our manuscript.

---

### Official Review · Reviewer_izi8 · 2023-08-10

**Soundness:** 4
**Typos Grammar Style And Presentation Improvements:** modal -> module [285 line]

**Excitement:**

4: Strong: This paper deepens the understanding of some phenomenon or lowers the barriers to an existing research direction.

**Paper Topic And Main Contributions:**

This paper addresses the challenging task of automatically determining the temporal context or date of historical texts, especially when explicit temporal mentions are not readily available.

Contributions of the paper:

1. Time-Aware Language Model (TALM): The paper introduces TALM, a specialized language model designed to capture temporal changes in word usage. Unlike traditional language models that often overlook diachronic variations, TALM leverages pre-trained models from general domains and adapts them to time-specific contexts. This enables the model to better understand and represent the evolution of language over time.

2. Hierarchical Document Modeling: The authors propose a hierarchical approach to model long diachronic documents. By encoding documents with temporal word representations obtained from TALM, the model captures implicit temporal information present in the text. This hierarchical representation facilitates the understanding of how language and topics change over extended periods.

**Questions For The Authors:**

It would be beneficial to mention in detail about the weight setting and parameters in more detail. It wasn't available in the paper so it's confusing

**Reasons To Accept:**

Strengths of the paper:

1. Empirical Validation: The paper's empirical evaluation on a Chinese diachronic corpus demonstrates the practical effectiveness of TALM. Its outperformance compared to existing state-of-the-art approaches in historical text dating underscores the practical utility of the proposed method, making it a valuable addition to the NLP toolkit.

2. Insightful Interpretation of Results: The authors provide insightful interpretations of the observed performance trends. For example, the explanation of the difficulty in predicting the Southern Dynasty Liang due to semantic similarities with other periods and the impact of the short duration of the Later Jin period adds valuable context to the results.

The discussion of the confusion matrix for the model's predictions, particularly focusing on the challenges of predicting the time period of the Southern Dynasty Liang, demonstrates a thorough analysis of the model's strengths and limitations. This kind of analysis provides a deeper understanding of where and why the model may struggle.

**Reasons To Reject:**

1. Ethical and Societal Implications: Although the paper does not discuss potential ethical or societal implications explicitly, the use of language models for historical text analysis could raise concerns about potential biases, misinterpretations, or controversial narratives. The paper should address these considerations to ensure responsible and unbiased historical analysis.

**Reproducibility:**

3: Could reproduce the results with some difficulty. The settings of parameters are underspecified or subjectively determined; the training/evaluation data are not widely available.

**Reviewer Confidence:**

4: Quite sure. I tried to check the important points carefully. It's unlikely, though conceivable, that I missed something that should affect my ratings.

---

> ### Author Rebuttal · Authors · 2023-08-28
>
> Q1: Ethical and Societal Implications: Although the paper does not discuss potential ethical or societal implications explicitly, the use of language models for historical text analysis could raise concerns about potential biases, misinterpretations, or controversial narratives. The paper should address these considerations to ensure responsible and unbiased historical analysis.
>
> A1: Thanks for your comment. We will consider the improvement to alleviate ethical and societal Implications to achieve an unbiased historical text analysis model.
>
> Q2: Limited Generalizability: The paper focuses on a specific historical context, the Chinese diachronic corpus. The generalizability of the proposed TALM model to other languages, cultures, or historical periods might be limited. This could potentially restrict the applicability and impact of the proposed approach beyond the Chinese historical context.
>
> A2: Thanks for your comments. The aim of this work is to explore the general method for historical context modeling. To this end, neither the research hypothesis on languages, cultures or historical periods is introduced, nor task-specific features are employed in our model. We think there are no explicit clues to show that there is the generalizability problem of our model. Despite that, we are willing to validate it by adding experiments of other datasets and report it in the revised version of our manuscript.
>
> Q3: It would be beneficial to mention in detail about the weight setting and parameters in more detail. It wasn't available in the paper so it's confusing.
>
> A3: Thanks for your comment. Details of weights and parameters are not given in the manuscript due to the page limitations. We will add them in the revised one.

---

### Official Review · Reviewer_WaJ5 · 2023-08-13

**Typos Grammar Style And Presentation Improvements:** NA
**Soundness:** 4

**Excitement:**

4: Strong: This paper deepens the understanding of some phenomenon or lowers the barriers to an existing research direction.

**Missing References:**

NA

**Paper Topic And Main Contributions:**

Summary: The paper proposes a time-aware language modeling approach for historical text dating. The authors argue that traditional language models are not well-suited for this task, as they do not take into account the temporal context of text. To address this, the authors propose a model that jointly models text and its timestamp. This allows the model to learn the temporal patterns of language, which can be used to improve the accuracy of text dating.

Novelty: The paper makes several novel contributions. First, the authors propose a novel time-aware language modeling approach for historical text dating. Second, the authors evaluate their approach on a large dataset of historical texts and show that it outperforms existing state-of-the-art models. Third, the authors provide a detailed analysis of the results, which helps to understand the strengths and weaknesses of their approach.

**Questions For The Authors:**

- The max precision of the proposed method is 84.96. What are the cases of False positives?


**Reasons To Accept:**

This is a technically sound paper with detailed evaluation and experimental clarifications. Below are the detailed descriptions:

Technical soundness: The paper is technically sound. The authors provide a clear and well-motivated explanation of their approach. The experimental evaluation is rigorous and the results are convincing.

Evaluation: The paper evaluates their approach on a large dataset of historical texts. The results show that their approach outperforms existing state-of-the-art models. The authors also provide a detailed analysis of the results, which helps to understand the strengths and weaknesses of their approach.

Clarity: The paper is well-written and easy to understand. The authors provide clear and concise explanations of the concepts and methods. The experimental results are also presented in a clear and informative way.
Significance: The paper makes a significant contribution to the field of natural language processing. The proposed approach has the potential to be used in a variety of applications, such as historical research and digital humanities.


**Reasons To Reject:**

1) The authors could provide more details about the dataset that they used to train their model.
2) The authors could also provide more details in the analysis sections, especially by showing the examples on where the proposed model fails.
3) The authors could explore the use meta temporal features, such as the author of the text and the genre of the text.


**Reproducibility:**

4: Could mostly reproduce the results, but there may be some variation because of sample variance or minor variations in their interpretation of the protocol or method.

**Reviewer Confidence:**

4: Quite sure. I tried to check the important points carefully. It's unlikely, though conceivable, that I missed something that should affect my ratings.

---

> ### Author Rebuttal · Authors · 2023-08-28
>
> Q1: The authors could provide more details about the dataset that they used to train their model.
>
> A1: Thanks for your comment. Full details of the dataset are not given in our manuscript due to the page limitations, and will be added in the revised version.
>
> Q2: The authors could also provide more details in the analysis sections, especially by showing the examples on where the proposed model fails.
>
> A2: Thanks for your comment. In the revised version of our manuscript, we will give more details about the error cases.
>
> Q3: The authors could explore the use meta temporal features, such as the author of the text and the genre of the text.
>
> A3: Thanks for your comment. There is no doubt that meta temporal features help to text dating, which has been discussed in the related work section. Such features, however, are not taken into account in this work, mainly because we hope that this work can be applied in some real application scenarios, such as dating a historical document without any meta information(it usually happens to ancient China manuscripts in archaeological findings). We are also happy to discuss the contribution of the meta information of a historical document when it is available.
>
> Q4: The max precision of the proposed method is 84.96. What are the cases of False positives?
>
> A4: In most cases, predicted results of false positives are temporally close to the ground truth, according to the confusion matrix in Figure 1. It matches the linguistic intuition that languages of adjacent time periods are more similar than those having a long temporal distance, which also brings the challenge of determining whether a historical text belongs to a time period or an adjacent one. We will show more details of the false positives in the revised manuscript.

---

### Meta-Review · Area_Chair_w86b · 2023-09-27

**Recommendation:** 4

**Metareview:**

Overall, the paper is acknowledged for its technical soundness, clear explanations, and insightful interpretation of results. The reviewers suggest providing more details about the dataset, exploring the use of meta-temporal features, addressing ethical and societal implications, and expanding the discussion to encompass other languages.

---

### Decision · Program_Chairs · 2023-10-07

**Decision:**

Accept-Findings

**Comment:**

Overall, the paper is acknowledged for its technical soundness, clear explanations, and insightful interpretation of results. The reviewers suggest providing more details about the dataset, exploring the use of meta-temporal features, addressing ethical and societal implications, and expanding the discussion to encompass other languages.